# A One-Dimensional Non-Intrusive and Privacy-Preserving Identification System for Households

Tomaž Kompara [1,2,*], Janez Perš [3,*], David Susič [1,*] and Matjaž Gams [1,2,*]

1 Department of Intelligent Systems, Jožef Stefan Institute, Jamova Cesta 39, 1000 Ljubljana, Slovenia
2 Jožef Stefan International Postgraduate School, Jamova Cesta 39, 1000 Ljubljana, Slovenia
3 Faculty of Electrical Engineering, University of Ljubljana, Tržaška Cesta 25, 1000 Ljubljana, Slovenia
* Correspondence: tomaz.kompara@ijs.si (T.K.); janez.pers@fe.uni-lj.si (J.P.); david.susic@ijs.si (D.S.); matjaz.gams@ijs.si (M.G.)

**Abstract:** In many ambient-intelligence applications, including intelligent homes and cities, awareness of an inhabitant's presence and identity is of great importance. Such an identification system should be non-intrusive and therefore seamless for the user, especially if our goal is ubiquitous and pervasive surveillance. However, due to privacy concerns and regulatory restrictions, such a system should also strive to preserve the user's privacy as much as possible. In this paper, a novel identification system is presented based on a network of laser sensors, each attached on top of the room entry. Its sensor modality, a one-dimensional depth sensor, was chosen with privacy in mind. Each sensor is mounted on the top of a doorway, facing towards the entrance, at an angle. This position allows acquiring the user's body shape while the user is crossing the doorway, and the classification is performed by classical machine learning methods. The system is non-intrusive, non-intrusive and preserves privacy—it omits specific user-sensitive information such as activity, facial expression or clothing. No video or audio data are required. The feasibility of such a system was tested on a nearly 4000-person, publicly available database of anthropometric measurements to analyze the relationships among accuracy, measured data and number of residents, while the evaluation of the system was conducted in a real-world scenario on 18 subjects. The evaluation was performed on a closed dataset with a 10-fold cross validation and showed 98.4% accuracy for all subjects. The accuracy for groups of five subjects averaged 99.1%. These results indicate that a network of one-dimensional depth sensors is suitable for the identification task with purposes such as surveillance and intelligent ambience.

**Keywords:** one-dimensional depth sensor; biometrics; identification; machine learning

## 1. Introduction

For many years, Artificial Intelligence (AI) has played a central role in techniques that improve system performance in various areas, especially when Machine Learning (ML) has been used. The immense growth of data due to the Internet and Internet of Things, as well as the increase in computing power, has led to a great increase in the benefits of AI in the last decade [1], as there are many data to learn from, and the amount and speed of data exceeds human capabilities. Therefore, AI has a significant impact on people's daily lives nowadays. Ambient Intelligence (AmI) enhances people's everyday lives by sensing their presence and responding to their actions [2,3]. To provide this service without interfering with the users' activities, the sensing must be non-intrusive. This means that the users perform their activities in exactly the same way as if the sensors did not exist. Furthermore, the sensing should preserve the users' privacy as far as possible, so that such systems can be used in private environments such as smart homes [4]. In addition, sensors should be robust, low cost and highly accurate to be used in real life situations. These are the main requirements for AmI sensors that need to be met in order to introduce them into everyday life.

The key information that allows ambience to adapt to the individual user is the number of people who are present and their identities [5]. This information is helpful in order to put additional sensor information into the proper context. For example, a user comes home and turns a thermostat to 20 °C. After a while, his wife comes home and sets the thermostat to 22 °C, since this is the temperature she is comfortable at. If the users' identities are provided, the AmI system can regulate the temperature of the room according to the preferences of the users who are present, without any further input. Similar problems affect cooling, ventilation, assistance for the elderly, security systems, etc. In the same way, non-essential functions, such as the choice of music, movies and even commercials, can be controlled based on the identity of the people who are present. Furthermore, the history of the inhabitants' presence can be used to predict their next action and adjust the ambient environment in advance (controlling robot cleaners, domestic hot-water preparation and powering standby devices) [6]. Therefore, detecting the number and identities of the people who are present is one of the preconditions for successful AmI applications. It should be noted that the identification task for setting room preferences is quite different in nature compared to the identification when entering a smart home. The inability to classify correctly an entry in a home can lead to severe consequences, whereas the smart home taking care of room residents can always set the preferences to default values if the person entering a room is not classified with sufficient probability.

Laser-based technology has made remarkable progress in recent decades. It covers a wide range of fields, such as medical sciences, space sciences and military technologies [7,8]. Laser sensors are capable of detecting, counting, imaging and scanning distances and proximity, making them ideal for numerous applications such as vehicle automation and guidance, traffic management, security and surveillance and warehouse management. They are therefore ideal when it comes to home sensor applications such as user silhouette detection and authentication.

In recent years, many non-intrusive identification systems were designed; however, most of them are not able to preserve privacy, while at the same time obtaining a high identification accuracy. This weakness was the motivation behind our research. For the sensor to be non-intrusive, no device has to be carried by the user for the identification to be successful, nor is any additional interaction required—a typical example of such needless interaction would be putting a finger on a fingerprint scanner. For practical purposes, this narrows the sensor selection down to measuring the biometrically relevant physical properties of people. In our case, the shape of the human body was selected as the main measure. To preserve privacy, a one-dimensional depth sensor mounted on the top of a doorway, looking downward at an angle, was used. In this way, the person's body shape is not captured with a single shot, but with multiple measurements during the whole doorway-crossing event. Effectively, the sensor follows the best practice for designing surveillance systems, as set forth by privacy regulators worldwide, known as Privacy by Design: the system should acquire only the essential data required to solve the problem it addresses. This means that such a system cannot provide more data than it needs, even in the event of a third-party intrusion; all the sensor obtains is a partial, one-sided and relatively low-resolution depth map of a person, and that is all that an attacker could possibly gain. If, for example, a live video feed were used to recognize people, this would mean potentially catastrophic privacy consequences in the event of an attack, especially if such devices are used in private environments. On the other hand, the classification accuracy should remain as high as possible even with such "blurred" data having in mind that in a private home only a few people are to be classified. Finally, we are not interested in exact indoor location to set the room ambient parameters. These are the main assumptions leading to our approach. In the following, formal definitions of being invasive, intrusive and privacy-preserving are introduced:

**Definition 1** (Invasive). *The term invasive in this paper encompasses involving entry into the living body, e.g., by an incision or insertion of an instrument, or similar entry into the mental or cognitive state, including the implantation of changes in the well-being or emotions [9].*

In plain English, to be invasive is to have something introduced into the user's physical or mental state, either voluntarily or involuntarily. Physically, there does not have to be an actual physical insertion into the body, for example anything that applies pressure to any part of the user's physical state is considered invasive since it changes the physical state at the point of pressure. Psychologically, it is a bit similar and a bit different in that no physical influence is required, but the cognitive or psychological effect is similar—if something is introduced (or pushed) into the user's mental state or behavior by some device or system, it is considered invasive in the non-physical sense.

**Definition 2** (Intrusive). *In this paper, the term intrusive refers to the disruption of user's normal behavior [10].*

By analogy to the definition of invasive, the term intrusive is physical, mental or both. In addition, the term intrusive is often relative and refers to the culturally accepted activities in a particular community. For example, if a person is to enter a home, a certain activity is required to pass a security test, such as unlocking a door with a door key. If such entry is generally accepted, it is considered non-intrusive. However, if a camera system is introduced that unlocks the door based on facial recognition when a user enters the department, this system is considered non-intrusive and the prior entry with the door key becomes intrusive as a user has to unlock the door compared to simply approaching the door when facial recognition is used. Another example of change would be the automatic unlocking of the door of some modern cars which happens when a user approaches with a key, which is obviously non-intrusive, and the current normal unlocking of the car door by pressing a button or using a car key now becomes intrusive.

This paper is about passing an unlocked interior door that is without doors, with opened doors or needs to be opened in some way. If a user passes the door as usual and no additional load is introduced by the AmI system, this is considered non-invasive and non-intrusive.

**Definition 3** (Privacy-preserving). *The term privacy-preserving in this paper refers to the concept of security or harmlessness of user data when the data are or could be transmitted or communicated between different parties. The other party is not able to draw a potentially harmful conclusion from the data obtained [11].*

An example of non-privacy-preserving data are images taken with a camera. Even without actual security issues, users are usually uncomfortable with the knowledge that some device is taking accurate pictures of them. On the other hand, if the image captured by the camera is blurry enough that no one can see anything potentially harmful from it, it is considered privacy-preserving. This term is—as the previous two terms—in contradiction with detection accuracy. Nevertheless, there are systems that are both privacy-preserving and sufficiently accurate—for example, a left–right blurred image caused by a mechanical lens might allow correct height detection. In reality, systems of different types each establish their own relationship between these properties and accuracy, trying to accommodate for user needs and preferences.

Our approach is based on two hypothesis:

**Hypothesis 1.** *Due to the advancements of the laser devices and AI, the proposed laser-based system using AI methods will enable highly accurate identification of a small number of residents in a typical home.*

**Hypothesis 2.** *The introduced system will be non-invasive, non-intrusive and privacy-preserving.*

The process of data collection is described in detail to show that Hypothesis 2 is indeed satisfied, while the measurements of classification accuracy reveal relationships among various factors that affect identification accuracy to show conditions under which Hypothesis 1 is valid.

Since no such device exists in the mass market, the task is quite challenging and involves another parameter: the cost scheme. The goal is to use a device that should not exceed $100 as a general threshold.

In the remainder of the paper, we first present related work. After the related work, we describe the preliminary study conducted using the publicly available anthropometric measurements database. Next, we describe our system setup and its geometry. We continue with a description of the feature extraction process that describes the user's body shape. We then describe how the extracted features are used to determine direction and identity, as well as the evaluation process, its results and the comparison between theoretical estimates from a database and practical measurements. Finally, we present a discussion and conclusions with pros and cons.

## 2. Related Work

In the past decade, we have witnessed many attempts to develop identification systems that could support AmI applications on a ubiquitous scale and that could be easily integrated into the environment itself.

The two main application requirements are a high identification accuracy and non-intrusiveness. The first requirement arises from the need to correctly identify a person in order to properly personalize the environment. Every misidentification could lead to a user's discomfort, security risks and/or non-optimal energy use. The second requirement allows the user to maintain his/her way of living and interaction with the environment in exactly the same way as if the identification system did not exist. To meet both requirements, different sensors and techniques were explored in the scientific community. The most promising non-intrusive identification methods are the following:

- Pressure sensors are installed in the floor and used for measuring the location and the force of a foot. The user has to step onto the sensed area where the sensor is installed to be identified. It has been demonstrated that people can be identified according to the force profile of their footsteps [12,13]. Orr and Abowd were able to achieve 93% precision using 15 test people and a variety of footwear. Middleton et al., on the other hand, used an array of binary contact sensors and achieved 80% precision using the same number of test people.
- Doppler-shift sensors can determine an object's velocity based on a reflected wave's frequency shift caused by a moving object. According to Kalgaonkar and Raj [14], this sensor can be used to identify users based on their walking signatures when they walk straight towards the sensor. They obtained 90% accuracy for 30 test subjects in a laboratory environment, where only a single subject was observed at a time.
- Cameras are the most widely explored and used non-intrusive identification sensors. They are used to identify people with both face and gait recognition. Face-recognition methods much depend on lighting, facial expression, rotation, number of training examples and similar parameters [15]. Reported precision values vary widely between 37% and 98% [16]. Gait-recognition methods are mostly based on a person's silhouette dynamics. An accuracy of 87% was achieved by Tao et al. [17] for a single person walking on different surfaces. This number falls dramatically to 33% when the view angles, shoe types and surface types are varied. In some environments, people's activities and motion can be used to recognize identity, as in [18], where an 82% accuracy is observed. There were also some attempts to use extremely low temporal and spatial resolution cameras; however, the collected data can also be used for activity recognition, which may be undesirable [19]. Recent work in this field focuses mostly on much more difficult re-identification problem (e.g., [20,21]). Recent approaches in this field are based on deep learning (e.g., [22–24]).

- Scanning range-finders emit a signal and determine the distance of an object according to the time of flight, phase shift, or reflected angle. Such sensors identify people according to their body dimensions, which is similar to our approach. By using range-finders only the distance of a single point, a line or a full two-dimensional depth map can be obtained. On the basis of multiple single-point sensors, a 90% accuracy on three test subjects was achieved by Hnat et al. [25], whereas a 94% accuracy was obtained on eight test subjects by Kouno et al. [26] using full, two-dimensional depth images, obtained with the widely available Microsoft Kinect sensor.
- A radar-based system mounted at the top of the doorway that analyzes the signals reflected back from the environment to perform the identification is described in [27]. The identification accuracy for a group of an eight people was 66%. Nonetheless, this approach is proven to be good at people presence estimation [28].
- The thorough description of human monitoring system based on WiFi connectivity, in a fashion strictly compliant with the Internet of Things (IoT) paradigm, is given in [29]. In [30], a human identification system that leverages unique fine-grained gait patterns from existing WiFi-enabled IoT is proposed. The system achieves an average human identification accuracy of 91% from a group of 20 people.
- In [31], height based approach using a thermal camera is presented. The authors reported 92.9% accuracy for people with more than 2.5 cm difference in height tested on 21 subjects.
- In [32], a system is described that exploits pulsing signals from the photoplethysmography sensor in wrist-worn wearables for continuous user authentication. The system relies on the uniqueness of the human cardiac system and requires little training for successful deployment. In combination with the location recognition methods already mentioned, this system shows great potential as it enables seamless user authentication. However, it is not entirely non-intrusive, as the user must carry a wrist-worn wearable.

The authors would like to draw the reader's attention to some other systems that deal with privacy and authentication based data collection. Mobile crowd sensing is a sensing paradigm that uses sensor data from mobile devices carried by humans from a crowd of participants. Such data aggregation systems need to ensure the privacy of each participant [33]. A novel approach that combats this problem very successfully is presented in [34]. The mobile crowd-sensing system is not directly related to our work, as it aggregates data from all participants and does not focus on authenticating each participant. Privacy-preserving authentication in information retrieval from geographically based social networks is described in [35]. This method differs from ours in that it depends on knowing the geographic locations of each end system contained in the network, whereas our approach identifies the user only when it passes through the door.

It is our belief—and a strongly expressed view of privacy regulators, especially in the EU—that a high identification accuracy and non-intrusiveness are not the only requirements that should be considered. Privacy invasiveness should also be taken seriously, since the risk–benefit ratio is an important aspect, both when users are deciding whether or not to use the system and when privacy regulators are deciding whether to allow it for a particular purpose or not (as the [36] states: "Member States shall provide that personal data must be ... adequate, relevant and not excessive in relation to the purposes for which they are collected and/or further processed..."). Sensor solutions for AmI are especially sensitive in this aspect, as they mainly provide higher comfort to the users, and therefore the privacy threshold is particularly high. Privacy regulators are bound to be more conservative when evaluating such solutions, unlike the applications where security or people's lives are at a stake.

Although all identification systems can be—and usually are—designed to prevent third-party access, there is always a security risk from an adversary attack, which might expose privacy-sensitive data. One of the most promising approaches to promote privacy and data-protection compliance from the start is Privacy by Design [37,38]. A key guideline

to reduce the privacy risk is through minimizing the amount of data, meaning only absolutely necessary information is acquired and used in the process, and only a bare minimum is stored.

From the identification system's perspective, the amount of acquired data should be adjusted according to the required identification accuracy and the purpose of use. For example, in high-security environments (e.g., banks and nuclear power plants), the identification accuracy should be as high as possible, regardless of the privacy invasiveness. On the other hand, in private environments, e.g. people's homes, where the identification system mostly controls entertainment electronics and other appliances, privacy has a higher priority than accuracy. Therefore, to make identification systems suitable for home use, the privacy issues need to be resolved. For example, cameras can be used not only for person identification at a doorway crossing, but for other purposes as well, such as recognizing users' activities, clothing, social behavior and interaction. According to the Privacy by Design approach, such an amount of acquired data is excessive for the identification task alone. Therefore, neither camera-based nor a two-dimensional depth-sensor-based approach is suitable for the identification task in private environments without further constraints on data acquisition.

On the other hand, unlike surveillance and security systems, many tasks of the intelligent environment do not require extreme positional accuracy. For example, tasks such as personalized adjustment of room temperature, lightning and (possibly) music volume do not require precise localization of people; this results in both less expensive solutions and greater privacy protection. Additionally, in home environments, the number of people is small and we assume they are cooperative, e.g., it is not in their interest to seek out faults in the system, but they may be highly sensitive to privacy issues. Our approach is intended for such cases.

Finally, the 1D laser scanner we use may be considered mature or even obsolete in terms of technology, but, again, this is not so if privacy concerns are a factor. In comparison to other sensors now widely used in home environment (e.g., Asus Xtion Pro and PrimeSense Carmine [39] provide 2D depth with accuracy about 4 mm in depth axis), our sensor can be seen simply as single scanline equivalent. It is important to understand that the latest methods using sensors such as Kinect v2 are able to infer very private information about people, such as facial expressions [40] and even comprehension [41]. Therefore, the moment such sensor is introduced into the home, it opens the door wide open to privacy violations.

The two approaches most similar to ours are found in [25,26]. Both rely on the subject's body shape for identification and both use depth sensors. However, the one in [25] only measures the depth at a single point, making it significantly less accurate, but, privacy-wise, it is very non-invasive. On the other hand, the one in [26] uses a Kinect sensor, obtaining a dense depth map. Although not used in [26], Kinect provides an RGB camera, which makes it a significant privacy risk, and a dense depth map has a significant field of view (therefore, enabling an adversary to observe additional activity in the area, which was not meant to be observed). In contrast to this, our approach provides high accuracy, but observes the environment only through a single, static, one-dimensional depth sensor and relies instead on the motion of the people across the doorway to acquire any useful information. Therefore, it cannot be used to observe the environment and the user's activity in any other way—except when a person moves through the doorway. Any other activity (e.g., a person standing and gesticulating) would result in a garbled depth map, unlike with 2D depth sensors or cameras.

## 3. Identifying People from Body Measurements

Related work shows that people can be identified to a certain degree even from small amount of information regarding their (apparent) shape. There are many approaches that use different sensor modalities, but they all rely on shape information:

- In some cases, researchers found that a single height measurement solves their problem; this could be done using scanning range-finders [25] or thermal camera [31].
- The next step in complexity is using a 2D sensor, e.g., camera, to obtain silhouette shape [20].
- A 2D sensor may be augmented with depth estimation (e.g., Kinect 3D sensor), which allows multiple anthropometric measurements from depth image data [21].
- Finally, ever more popular deep learning approaches actually learn which features they will use [24]. Algorithms are opaque, but we may assume that shape, whenever distinct enough for (re)identification, will be included at some point in the computation hierarchy.

Our approach assumes that the subjects can be identified using a limited number of body measurements of limited quality, as this intrinsically safeguards privacy. Before building the actual people-identification system, we examined the theoretical limits with regards to the accuracy of body-shape-based identification systems.

For this part of the study, we used the ANSUR database [42], which has 132 anthropometric measurements for 3982 subjects aged between 17 and 51. The measurements in the database were obtained manually with an accuracy in the millimeter range. However, the data provided by the depth sensor cannot achieve such high-level accuracy, especially in real-life environments. The main sources of noise for the measurement are different ways of dressing, different hair styles, body-weight variations, the objects being carried, etc. To simulate these influences on the identification accuracy, a uniform noise was added to the anthropometric measurements.

The relation among the peak-to-peak noise amplitude $i$, the number of people $N$ and the identification accuracy $E$ based on a single measure can be estimated using Equation (1). The measurement distribution $p$ was obtained from the ANSUR database.

$$E(i) = \int_{-\infty}^{+\infty} 1 - \left( \int_{x-i/2}^{x+i/2} p(y)dy \right)^N dx \qquad (1)$$

However, the computational complexity increases significantly with a growing number of subjects and the number of measurement combinations, which prevented us from an in-depth analysis of the identification accuracy for larger sets of measurements. Therefore, we decided to perform a large-scale simulation experiment based on randomly picking two, three and more subjects from the ANSUR database and identifying them. We chose four measurements that can be reliably obtained by our sensor: height, height of the shoulders, shoulder width and head width, as part of the silhouette. Based on those four measurements, the identification accuracy was estimated in the presence of a varying amount of noise.

The accuracy was calculated based on one million randomly chosen groups of 2–10 subjects. The chosen amount of noise was added to their measurements. The result, i.e., the recognition accuracy, which depends on the number of subjects and the amount of noise, is shown in Figure 1. Each point on the grid on the surface in Figure 1 represents one million random trials. The nearest neighbor algorithm [43] was used to determine the identity of the noised data. If the classified identity corresponded to the ground truth from the database, it was considered as correct; otherwise, it was considered as a false identification. In addition to using all four chosen measurements, we performed tests on the subsets of those measurements. In Figure 1, the identification accuracy of all four measurements, the best-performing combination of two measurements and the best single measurement are presented. The single measurement results were additionally verified using Equation (1) and provided nearly identical results.

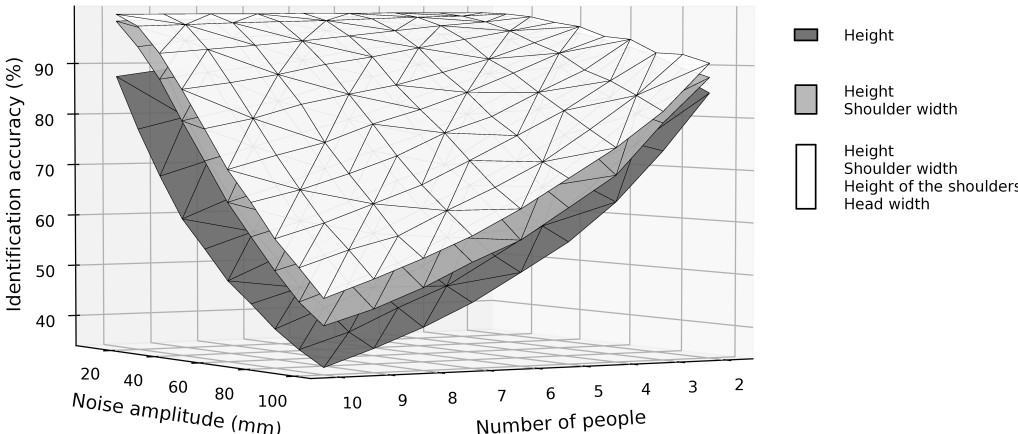

**Figure 1.** Theoretical identification accuracy according to the different measures, number of people and peak-to-peak noise amplitudes.

This analysis shows that combining more measurements increases the identification accuracy and can counteract the effects of noise, which significantly degrades the accuracy. Combining more measurements also allows us to identify subjects in larger groups, since the accuracy decreases with an increasing number of subjects as well. Note that this simulation was performed on four selected measurements that can be acquired with our sensor, while the developed one-dimensional depth sensor is capable of acquiring measurements of the whole body, i.e., not only shoulder width, for example, but also other widths, e.g., of the hips. On the other hand, Figure 1 indicates theoretic measurements with real-life obstacles such as dress or carried objects will decrease identification accuracy, which might make the identification useless in particular for larger groups of people. For a typical family, however, the number of family members is rather small, family members are usually of different heights and combining more measurements increases the overall accuracy, therefore Figure 1 provides an indication that our approach might be promising for home applications.

## 4. Sensor-Based Data Extraction

In this section, we describe the sensor device, the process of capturing data, identification and determination of the crossing direction from one-dimensional depth-sensor data, along with the sensor geometry and the overall system architecture. The transformation of the sensor data is applied to ensure data normalization, i.e., to ensure the compatibility of the acquired data between the sensors, regardless of the sensor characteristics and the mounting position. After the transformation procedure, the features are extracted and passed to the crossing-direction detection and the people-identification methods. While the basic approach presented here is not particularly novel, its understanding is important to properly describe the silhouette creation.

### 4.1. Sensor Geometry

The sensor is mounted on the top of the doorway facing down at an angle, as shown in Figure 2. Its mounting position allows the acquisition of one side of the user's body in a single doorway-crossing event. To ensure good-quality data, the sensor has to be set properly, as follows.

As shown in Figure 2, the sensor effectively projects a laser plane, which, by intersecting with the object of interest, forms a line. The plane's slope ($\alpha$ in Figure 2) depends on the sensor's mounting angle. Therefore, the angle $\alpha$ determines the size of the acquisition area. A smaller $\alpha$ results in a larger measuring area, and vice versa. A small measuring area is preferred, since only the people passing should be measured to preserve the privacy of other users, who may be present in a room, but are currently not passing through the doorway. However, with a smaller measuring are, a a higher sampling frequency is

required to obtain sufficiently detailed data. For that reason, the sensor's mounting angle should be chosen according to the sensor's sampling frequency.

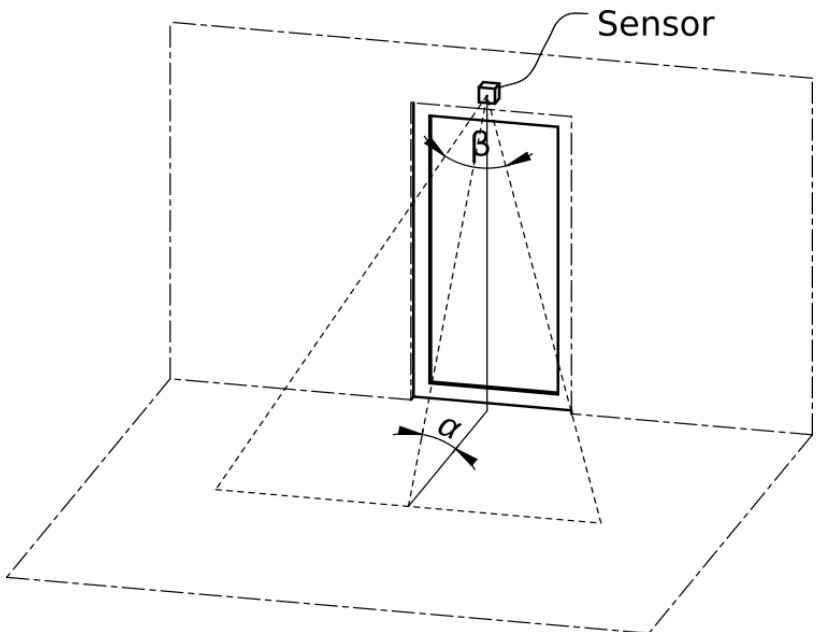

**Figure 2.** One-dimensional depth sensor mounted on the top of the doorway.

Other sensor properties that affect the quality of the measurements are the field of view ($\beta$ in Figure 2), the angular resolution and the raw sensor's accuracy. The field of view should, ideally, be 180° to cover the whole doorway area; however, smaller angles might be acceptable, depending on the size of the blind spots. The angular resolution is defined by the number of measurements along the field of view. From the acquisition perspective, $\alpha$ and the sampling frequency determine the vertical sampling resolution; $\beta$ and the angular resolution determine the horizontal sampling resolution; and the raw sensor's accuracy determines the depth resolution. All of these parameters can be changed to provide different compromises between privacy and performance, especially if the number of users is small and excessive identification accuracy is not needed. The architecture of a physical sensor is cost-effective—the material costs for the prototype were under \$100. It should also be noted that similar lasers are used in several home devices, including for small children to play with, as they are invisible to the human eye and pose no danger to humans or pets.

### 4.2. Self-Calibration and Input-Data Transformation

The transformation step is required to map the raw sensor data into the real-world Cartesian coordinate system. This enables the measured data to be used across different systems (e.g., across multiple sensors in the same household), regardless of the sensor type and its properties. The raw sensor data can be presented as a vector, where each value represents a distance at a specific angle (Figure 3, left). These data can be transformed into a coordinate system, as shown on the right of Figure 3.

The system can self-calibrate when no one is present. This can be done during the installation and repeated at any subsequent point, if needed. For self-calibration, we assume that the angle $\alpha$ is known and is fixed during the sensor's manufacturing. The sensor's height is then calculated using Equation (4), where $\alpha$ is the sensor's mounting angle and $L_{\beta/2}$ is the measured distance at the projected line midpoint, i.e., at $\beta/2$. The acquired depth with no one present is also recorded during the self-calibration stage to help with the users' body extraction at the acquisition stage.

During the acquisition process, all three Cartesian coordinates can be calculated using (3)–(5), where $L$ is the measured distance, $\beta'$ is the angle at which the distance was measured and $\beta$ is the field of view.

$$Y_0 = L_{\beta/2} \cdot sin(\alpha) \tag{2}$$

$$x = L \cdot sin(\alpha) \cdot cos(\beta' - \beta/2) \tag{3}$$

$$y = Y_0 - L \cdot sin(\alpha) \cdot cos(\beta' - \beta/2) \tag{4}$$

$$z = L \cdot cos(\alpha) \cdot cos(\beta' - \beta/2) \tag{5}$$

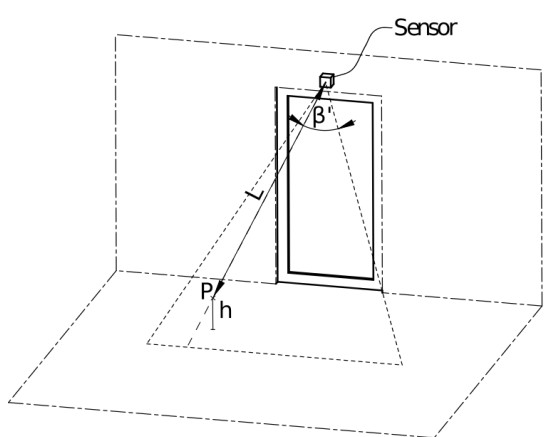 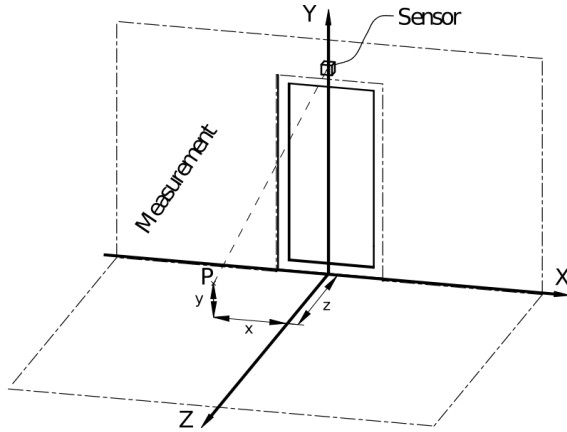

**Figure 3.** Transformation of the input data: (**left**) the depth measured with a one-dimensional depth sensor; and (**right**) the data transformed into the Cartesian coordinate system.

This kind of data transformation is valid, regardless of the type, technology and characteristics of a one-dimensional depth sensor. This allows the measured data or identification models to be shared between multiple sensors (e.g., in the same household). Sharing data through multiple identification sensors can be used to speed up the learning and consequently improve the accuracy. Therefore, with a data-transformation procedure, the identification accuracy of the whole sensor network can be improved.

### 4.3. Extraction of Features

To robustly identify the users, their body features need to be extracted from the transformed data. The measurements are pre-processed to remove the static background (e.g., ground and walls), leaving only measurements of the users for further processing. Next, for each non-background curve, five features are calculated. These features are used to determine users' crossing direction and identity.

The static background is removed from the data across the whole field of view by comparing the distance $L$ of each measurement with the corresponding ground measurement, obtained during the self-calibration stage. Unless the measurement is significantly closer to the sensor than the background measurement, that data point is discarded. In this way, we obtain a *non-background curve*.

Next, five features are extracted from each non-background curve (Figure 4). The first feature (6) is the horizontal ($x$) distance between the first and the last curve's point, representing the measured object's width. The second feature (7) is the horizontal surface area ($xz$) between the curve and the shortest distance between the first and and last point, roughly corresponding to the measured object's volume. The third feature (8) is the maximum perpendicular distance from the shortest line from the first to the last point and the curve, representing the maximum object curvature. The last two features (9) and (10) are the maximum measured height and its horizontal position, which provides the user's location. In this way, each non-background curve is represented with these five features, which describe the user's body shape at a measured position.

$$F_1 = |x_0 - x_n| \tag{6}$$

$$F_2 = \sum_{i=1}^{n} (x_i - x_{i-1})| * |\frac{z_i - z_{i-1}}{2} - \frac{(z_n - z_0)(x_i - x_{i-1})}{2(x_n - x_0)} \tag{7}$$

$$F_3 = max\left(\frac{|(z_n - z_0)x_i - (x_n - x_0)z_i + x_n y_0 - z_n x_0|}{\sqrt{(z_n - z_0)^2 + (x_n - x_0)^2}}\right); \tag{8}$$

$$\text{for } 0 < i < n$$

$$F_4 = max(y_i); \text{ for } 0 \leq i \leq n \tag{9}$$

$$F_5 = x(F_4) \tag{10}$$

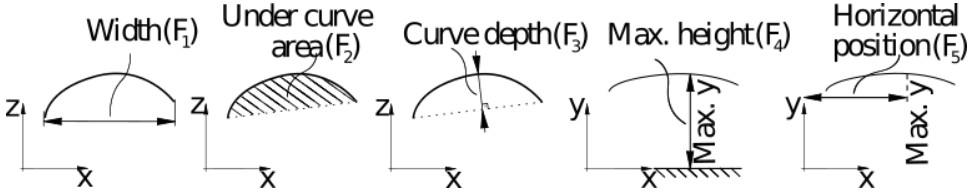

**Figure 4.** Graphical representation of the extracted features.

### 4.4. Acquisition Triggering and the Crossing Direction

The acquisition procedure is triggered when a non-background measurement (i.e., distances for the whole field of view) appears. The procedure stops when non-background measurements stop appearing. The acquisition yields a set of non-background measurements, which can represent one or more people crossing. To determine the number of people crossing between two background measurements, one or more rapid changes in the maximum height and/or horizontal position of the maximum-height features are observed, as shown in Figure 5D. A sudden drop and rise of maximum height feature indicates a new person is being measured. According to this logic, the input data are split into single person crossing events, which are later used to determine crossing direction and people identification.

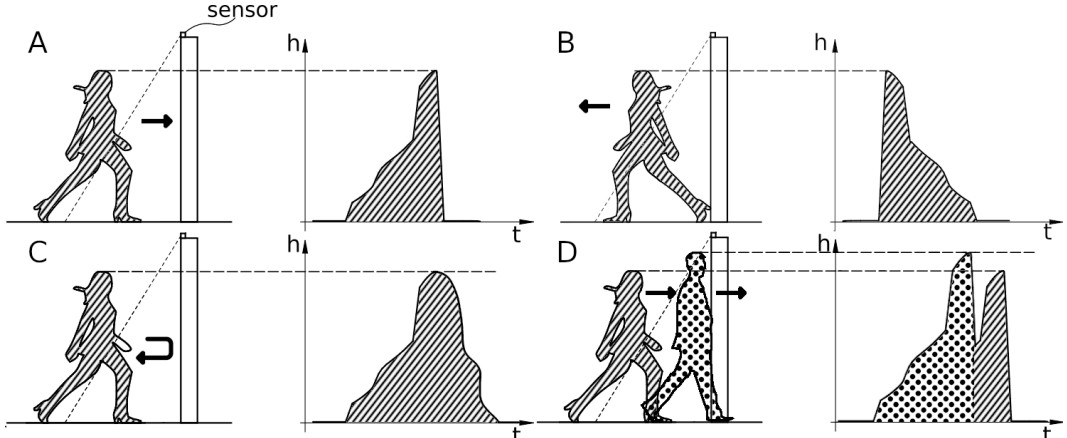

**Figure 5.** Response of the maximum-height feature (9) according to the crossing direction: (**A**) user walks toward the sensor; (**B**) user walks away from the sensor; (**C**) user first walks toward the sensor and then turns and walks away from the sensor; and (**D**) two users walk toward the sensor.

The crossing direction can be determined by observing the maximum-height feature. For each user, we keep a track of the maximum-height feature through the whole passing event. Due to the slope of the laser plane, the crossing direction can be determined, as shown in Figure 5. If the height rapidly increases and then slowly decreases (the laser beam

encounters person's head first and the heels last), the user walks away from the sensor, as shown in Figure 5B; otherwise, the user walks towards the sensor. In this manner, both walking directions as well as other combinations (i.e., the user starts entering and then exits and the other way around) can be determined.

*4.5. Identification*

People identification is done at every crossing event. If multiple people are detected, it is performed once for each person. The identification is based on the processed body features acquired from a doorway crossing event, as shown in Figure 4.

In our framework, we do not assume a constant velocity of people; moreover, we allow for variations in the direction (e.g., as a consequence of hesitation). Therefore, the maximum-height feature (9) is used to re-order other acquired features (i.e., width (6), area under curve (7) and the curve depth (8)) along the temporal axis in the descending order of the maximum height. An example of these features is shown in Figure 6. The horizontal position feature (10) is not used in the identification, since it does not carry any relevant identity information. With sorted features, even if a user stops, moves slightly backwards and then continues the crossing, the features are still valid.

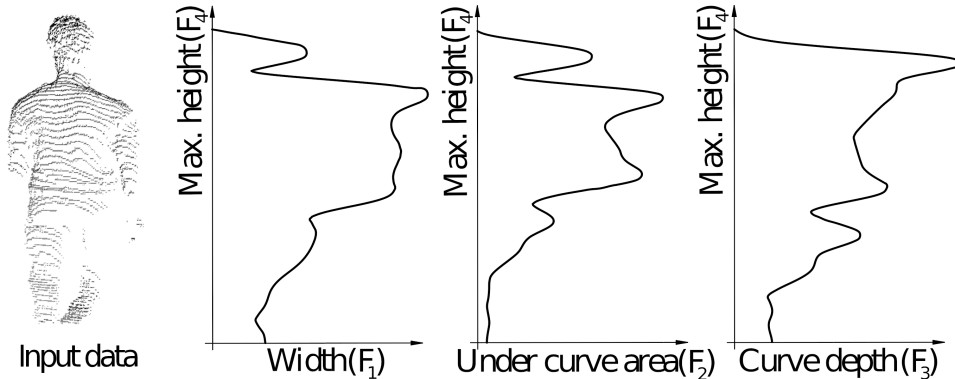

**Figure 6.** An example of input data and features used: width (6), curve depth (7) and area under the curve (8) sorted by the maximum-height feature (9).

Differences in the crossing velocity, which yield different amounts of data, are addressed by re-sampling the ordered features to the fixed number of samples. After re-sampling, a Fast Fourier Transformation (FFT) is used on each of the three features to obtain the Fourier descriptors (FD) [44]. From each re-sampled feature, 10 FDs are extracted (11). The maximum measured height is added to the 30 FDs, which gives 31 descriptors in total.

$$FD_{f,i} = |*| \frac{FFT(F_f, i)}{FFT(F_f, 0)}; \text{ for } 1 \leq f \leq 3 \text{ and } 1 \leq i \leq 10 \tag{11}$$

Finally, a ML algorithm is applied to the descriptors to determine the user's identity. Since only one side of the body can be measured in a single crossing event (using one sensor), front and back ML models have to be built. The appropriate module is selected based on the user's walking direction. If the user walks away from the sensor, we assume that the back of the body is measured; otherwise, a model for the front of the body is used. The necessary assumption here is that the user is walking upright and forward.

## 5. Dataset

To evaluate the identification accuracy of the proposed approach, we used the described sensor setup to acquire a dataset containing the data from 18 subjects. For this task, a new, one-dimensional depth sensor prototype was developed specifically to meet all the requirements of our approach. The prototype sensor consists of Raspberry Pi B+ (Raspberry Pi Foundation, Cambridge, UK), NoIR camera module V1.3 (Raspberry Pi

Foundation, Cambridge, UK), 110° 780 nm laser line module H9378050L (Egismos, Burnaby, BC, Canada) and Acrylic NIR Longpass Filter AC760 (Midopt, Palatine, IL, USA). Laser module includes optics which transforms the beam into the 2D plane, thus enabling depth measurements along the line and ensuring safe intensity of laser light. However, any sensor fulfilling the application requirements can be used, regardless of the technology employed, e.g., a laser scanning device.

The developed one-dimensional depth sensor is based on a triangulation method with a depth-measuring range of between 0 and 3 m. The depth resolution depends on the measured depth and varies from 0 to 50 mm, as shown in Figure 7. An average depth of crossing in a real-life application depends on the height of the installation of the sensor and the height of an entering person—an adult comes closer to the sensor compared to a small child. However, in an average application and for an average-height person, the depth of the most important body parts such as head, shoulders and hips (the last one often covered by hands) should enable around 10 mm resolution, while for head it should only be around 5 mm on average. The sensor-sampling frequency is 30 Hz where 1240 measurements are taken simultaneously across the whole 120° field of view. These characteristics allow us to obtain a sufficient amount of data in a single doorway-crossing event (Figure 6). The sensor was mounted at an angle $\alpha = 60°$, and all the results reported were acquired with that geometry. Despite the relatively high resolution, the privacy-accuracy compromise could be adjusted simply by changing $\alpha$.

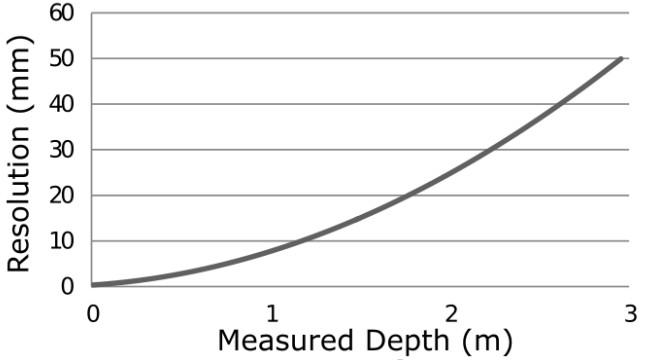

**Figure 7.** Sensor resolution with respect to the measured depth.

For the crossing direction detection and identification-accuracy evaluation, 670 crossing events involving 18 subjects aged between 21 and 47 were recorded. The identity and crossing direction of the subject in each passing event were recorded manually by an operator. The height of the smallest subject was 1.60 m and the height of the tallest was 1.94 m, whereas the average height of all 18 subjects was 1.76 m. The sample included 7 females and 11 males. The participants, all of whom were our colleagues at the institute, voluntarily signed written consent forms to conduct this research. At no time during the research was the health of the participants at risk since they only walked through a door and only a household common-type laser was applied on them. Both the privacy of the subjects and the confidentiality of the recorded data were maintained.

To obtain real-life scenarios, users' shoes and clothes (from a T-shirt to a jacket and a coat) were changed between the measurements. In addition, people carried everyday objects, such as umbrellas or backpacks, to make the conditions more realistic. In this way, a wide range of everyday scenarios was covered.

## 6. Experiments and Evaluation

To estimate the accuracy, a closed-set identification framework was used, i.e., the user was classified into one of the previously known identities representing, for example, the occupants of a household. This scenario is realistic for home use, where the number of occupants in a typical household is more-or-less fixed and adding another member is not a frequent event.

Overall, several experiments were performed to establish various properties of the proposed system in numerous scenarios. A couple of measurements are presented here. To determine how much data are needed to reliably identify people, two different approaches were explored. In the first approach, only the user's height was used, which is very privacy-preserving and easy to obtain. This approach is well known [45–47], and therefore served as our baseline. In the second approach, one-dimensional scanner data were used within our proposed framework, which might sound a bit privacy-invasive, but in reality it is not, and the additional data are essential to increase the identification accuracy. Both approaches were evaluated on the same dataset, but different amounts of acquired data were used to show the difference in the identification accuracy—only the maximum height in the first case and the full set of proposed descriptors when evaluating the proposed approach in the second case. In addition, another measurement was performed to evaluate which ML method performs best, but on a slightly different scenario.

### 6.1. Crossing Direction Detection

As described in the Acquisition Triggering and the Crossing Direction section, all the input data were split into single passing events by observing the maximum-height feature through the whole data acquisition process and then classified as if the person is walking in or out of the room. All 670 crossing events were classified correctly. The recorded crossings events were common for a home environment, i.e., only one person entering or leaving at the same time.

### 6.2. Identification by Subject's Height

It has been shown [45–47] that height can be used for the identification of smaller groups of people, whereas it is not suitable for larger groups, since sooner or later two users will be of equal height at least within the sensor's accuracy and under real-life circumstances. It should be noted that, in a real-life scenario, the measured user's height depends not only on the user's body measures but also on the shoes, haircut, head position and the speed or manner of walking. This makes an identification based on a single height measurement even harder and less robust in real life.

In the first experiment, the user's height is defined as the maximum height measured across a single doorway-crossing event (which is in our case $max(F_4)$). The distribution of measured heights is shown in Figure 8. Note that these measures could be obtained using single-point depth sensors, but we reused our depth data to calculate the subject's height for the purpose of this experiment.

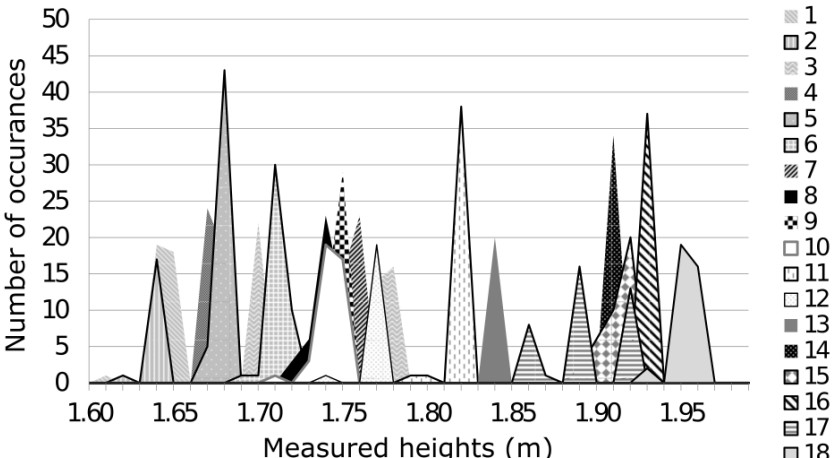

**Figure 8.** Distribution of heights measurements for the baseline experiment—identification by the subject's height. Different colors/textures denote different subjects.

The accuracy of the closed-set identification system based on height measures was estimated by using a nearest neighbor ML algorithm [43] and 10-fold cross validation [48]. The nearest neighbor algorithm classifies the person to the most similar person in the training set. To estimate how the size of a group affects the identification accuracy, the dataset was split into all possible combinations of groups of people. Next, each group of people was separately classified. The minimum, maximum and average closed-set identification accuracy with respect to the size of a group is presented in Figure 9 (lower curve). It is clear that the identification accuracy decreases with an increasing group size. Similarly, the maximum accuracy decreases, since larger groups of people are not diverse enough to properly identify all of the people with such a simple descriptor.

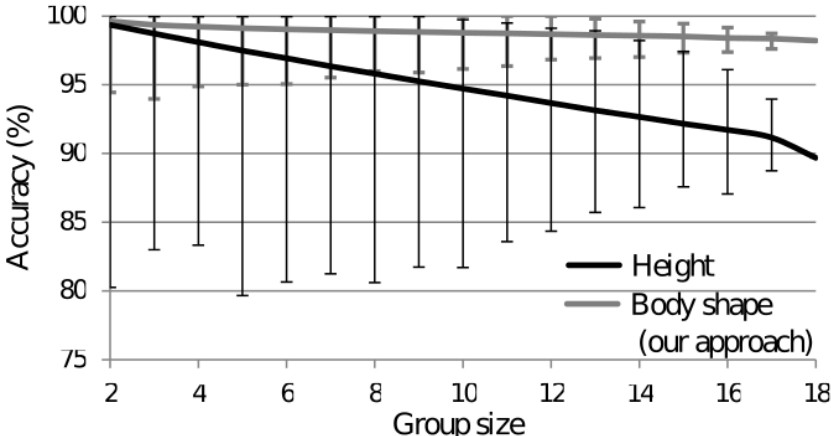

**Figure 9.** The minimum, maximum and average closed-set identification accuracy with respect to the size of a group. Whiskers denote the minimum and maximum value, as obtained by a 10-fold cross validation. The evaluation was made for different group sizes, from 2 to 18, taking into the account all the possible ways to construct groups of a certain size.

The results show that the closed-set identification accuracy using only the subject's height for all 18 subjects is 90%, which is quite better than described in [25], where the same accuracy was obtained for only three test subjects, but still not good enough for commercial use. We speculate that there might be several reasons for the relatively good results, e.g., not only better accuracy or position of the sensor, better sampling frequency or better height calculation, but also more consistent head cover, hair style or different shoe soles, as well as quite different heights of our randomly chosen test subjects. A much higher identification accuracy might not be achievable only by improving the sensor characteristics, since most of the noise comes from the environment. That is the rationale behind our approach—that AI methods are needed on redundant body measures to improve accuracy.

*6.3. Our Approach: Identification by Subject's Body Shape*

In this case, the input data are represented by 31 descriptors, as described in the Sensor-based Data Extraction section. Similar to the previous experiment, the closed-set identification accuracy estimation was made by using ML algorithms and a 10-fold cross validation for 18 subjects. Because of the larger number of descriptors, we tested several ML algorithms on the full dataset to establish the most appropriate one. The comparison of ML algorithms is shown in Table 1. A couple of the algorithms achieved similar very good performance, therefore there was no urgent need for testing or designing more algorithms at this stage, when the emphasis was on the sensor and the use of it. The best results were achieved by using the AdaBoostM1 ML algorithm [49], but, in terms of transparency, by a C4.5 classifier [50,51]—J48. Since a C4.5 classifier achieved such a good accuracy, it suggests that the 18 test subject were perfectly identified by the 31 descriptors if all data were captured and that additional noise such as object carried left enough descriptors intact to enable near-perfect identification. Using a full set of 18 subjects, AdaBoostM1

algorithm achieved 98.4% accuracy and is therefore our algorithm of choice. J48 (C4.5) is also an interesting choice particularly due to its simplicity, transparency and small number of required learning data, which is relevant for real-life applications. In the case of complex real-life domains, it might be more reasonable to opt for AdaBoostM1 since it usually achieves significantly better results than C4.5 [52]. Since Naive Bayes achieved only 87.76% accuracy, it indicates that the viewpoint through each single descriptor and then combined is less informative than the viewpoint through all of the descriptors combined at once.

**Table 1.** Comparison of ML algorithms in a 10-fold cross validation for 18 subjects.

| Classifier | Precision | Recall | F-Measure | Accuracy [%] |
|---|---|---|---|---|
| AdaBoostM1 (J48) | 0.98 | 0.98 | 0.98 | 98.36 |
| LogitBoost (DecisionStump) | 0.98 | 0.98 | 0.98 | 98.36 |
| Bagging (J48) | 0.97 | 0.97 | 0.97 | 97.31 |
| LMT | 0.97 | 0.97 | 0.97 | 97.16 |
| IB1 | 0.97 | 0.97 | 0.97 | 96.87 |
| J48 | 0.97 | 0.97 | 0.97 | 96.72 |
| SimpleLogistic | 0.97 | 0.97 | 0.97 | 96.57 |
| RandomForest | 0.96 | 0.96 | 0.96 | 96.12 |
| BayesNet | 0.95 | 0.94 | 0.94 | 94.33 |
| MultilayerPerceptron | 0.94 | 0.93 | 0.93 | 93.28 |
| RandomTree | 0.93 | 0.93 | 0.93 | 92.69 |
| LibSVM | 0.94 | 0.91 | 0.92 | 90.60 |
| NaiveBayes | 0.88 | 0.88 | 0.88 | 87.76 |

The results show a significant improvement in the average identification accuracy and its distribution compared to the height-only measurements, as shown by the minimum and maximum accuracy in Figure 9 (upper curve). The additional body-shape data in most of the cases enabled proper distinction of people of the similar height, which was the main purpose of using our approach. Furthermore, the narrower area of the identification-accuracy distribution suggests that this sensor is more reliable and robust than a sensor based on the user's height only.

*6.4. Comparison of Theoretical and Practical Measurements*

In Section 3, the theoretically computed accuracy under assumed noise and number of inhabitants is presented in Figure 1 to provide an initial estimate of which accuracy will be achieved under what circumstances when using a practical sensor setup. Moreover, the theoretical estimation allows an analysis on almost 4000 individuals and not only on 18 as in the physical measurements.

On the other hand, the practical measurements could show significantly different characteristics than the theoretical measurements, so a comparison between the two approaches is needed. In Figure 10, the experimental results marked with "our approach" are related to those from the simulation (without "our approach"). There are some similarities and some differences. In both approaches, whether theoretical or practical measurements, more body measurements improved the accuracy. When the same body measurements were compared, the theoretical results were slightly worse than the practical ones. The obvious reason for this is that there were 18 subjects with specific body sizes and nearly 4000 subjects in the database where, for example, many body sizes were similar and some were indistinguishably similar. As a result, the height comparison of the two approaches differs the most. Classification using only two physical measurements, e.g., height and shoulder width, already provided enough differences in the population to reduce the difference in accuracy between the physical and theoretical results. This is because the number of individuals with similar height is much smaller than the number of individuals with similar height and similar shoulder width. Since none of the theoretical measurements includes the full silhouette as in our approach, it was expected that the results of the

practical measurements would be slightly better than the theoretical evaluations of the database, and indeed the comparison confirms this. However, the inclusion of four body measurements allowed an accuracy of about 97% for 10 inhabitants even in the theoretical measurements. In summary, the theoretical and practical measurements are sufficiently compatible to confirm the first hypothesis of our approach, once on 18 and the second time on almost 4000 individuals, from which combinations of 2–10 individuals were selected for the experiments.

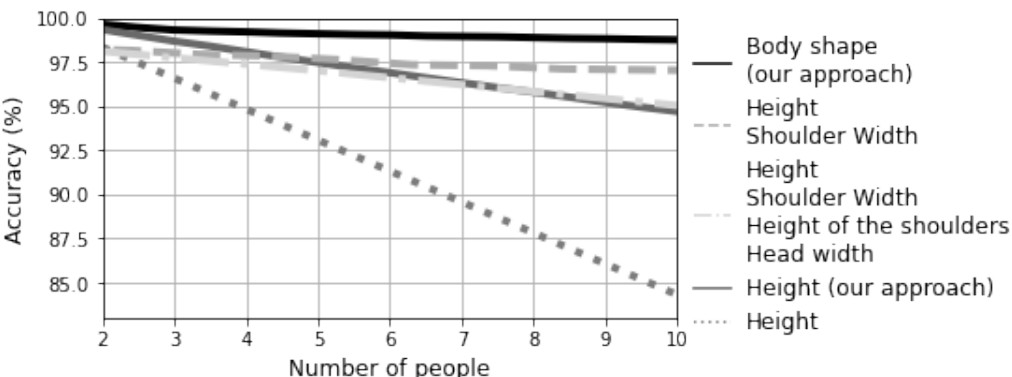

**Figure 10.** Experimental results along with theoretical identification accuracy according to different measures.

## 7. Discussion and Conclusions

In this paper, we present and test a sensor based on one-dimensional measurement for a novel approach to non-intrusive, non-invasive and privacy-preserving people identification in a smart environment, e.g., a household. Since the required accuracy in this AmI application is not as high as in security applications, and the number of people in a household is typically small, the proposed approach based only on body height or silhouette, e.g., body features, may prove sufficient for real-world applications.

Our results show that even the height-only approach achieves about 95% accuracy for nine tested individuals. However, the classification accuracy decreases rapidly as the number of people increases, dropping to 90% for 18 people, which means one error for 10 entries, which seems too high.

When the body shape approach is used instead of simple sensors that only measure body height, which requires the introduction of AI methods and more advanced computing, the accuracy is much higher and the results are more consistent (see Figure 9). The accuracy for 18 individuals remains above 98%, i.e., two errors in 100 entries. For five individuals, the accuracy reaches 99.1%. At the same time, the privacy intrusion remains practically nonexistent. Moreover, the ability to determine the crossing direction enables knowledge of the person's location down to the room level, as demonstrated in our experiments. With a cost-effective sensor implementation in terms of hardware and software, the laser-based approach, especially the body shape-oriented one, seems to be well suited for smart environments where data about the presence of people is needed but privacy must be maintained to a high degree. However, for high precision performance, practical application seems to be limited to families and small businesses with a small number of users.

The proposed identification system was evaluated in the laboratory on an apparent real-world scenario with 18 subjects, and the results were consistent with the preliminary study conducted using the ANSUR database. The live tests with our method on 18 subjects and the tests simulated with the ANSUR database on one million randomly selected groups of 3982 subjects consistently show that sufficiently good classifications can be obtained with the characteristic anthropomorphic data, e.g., height, shoulder width and hip width, for a small number of residents and under a reasonable noise. On the other hand, randomly

selecting a small test group from the pool of nearly 4000 subjects more often results in individuals with similar characteristics, especially when the number of descriptors is small, e.g., height only. When multiple descriptors are considered, the differences between the practical experiments in the laboratory and the theoretical classifications from the ANSUR database fade away.

To further improve the detection performance, misidentified crossing events were manually checked. It was found that the sensor had problems with depth measurement in the presence of reflective clothing and different hairstyles. This problem could be addressed with an improved sensor design, which should then improve identification accuracy, or through improved AI methods such as Deep Neural Networks (DNNs). Nevertheless, both the worst and average identification accuracies remained high despite the depth measurement errors. Secondly, it was found that interference from another light source, such as sun reflections on the ground, made the results unreliable. However, this seems to be a problem for the type of sensor used in the experiments and can be avoided by using a different type of sensor or with additional filters.

For further work, advanced ML methods along with new feature extraction are being considered, including DNNs. However, such methods are not applicable in real life since the learning time of the system is short. Newer methods enable fast learning even with DNNs, but these new approaches need to be implemented and tested in detail. In addition, new methods for measuring one-dimensional depth are being tested to improve the overall identification accuracy. As shown in the preliminary study, the higher is the number and quality of features, the better is the identification accuracy. With further improvements, we believe it is possible to achieve face-recognition accuracy without violating privacy. In addition, extensive measurements are planned to test the performance with multiple people entering the room.

Finally, we present pros and cons of the proposed approach. The experiments strongly suggest that the laser-AI approach enables decent accuracy for real-world applications within the $100 target for a sensor; the system is reliable, non-intrusive, non-invasive and does not compromise user privacy. In addition, there is no comparable market solution to date.

However, in the cons category, there are a couple of obstacles. First, with five rooms in the department, it already requires $500, which can be somewhat of a dilemma. With massive use, the cost might go down to even $400, but it is yet to be achieved. In addition, each of the sensors requires power and either cables or batteries provide additional nuance. There are also competing solutions that are significantly cheaper than the laser system. An example would be a computer camera that fits into the $10 category. Such an approach introduces privacy concerns, as observed in several surveys, but there is still a certain percentage of users who trust their security systems to prevent capturing personal data for third parties. Finally, the benefit of knowing who is in a room may not be essential for ordinary residents, even though it is often declared as such in AmI publications. Most people either live alone or in fixed combinations in their own rooms such as bedrooms, and there the settings can be permanently matched with any motion sensor to alert the smart home that the room is no longer empty.

In summary, our approach based on a laser sensor and AI software enables sufficiently good accuracy to be used in real life, especially if users prefer non-invasive, non-intrusive and privacy-preserving systems for their smart home. The biggest concern is whether the cost–benefit ratio is indeed beneficial for current smart homes.

**Author Contributions:** Conceptualization, T.K. and J.P.; methodology, T.K. and J.P.; formal analysis, T.K. and J.P.; data curation, T.K. and J.P.; writing—original draft preparation, T.K. and J.P.; writing—review and editing, D.S. and M.G.; and supervision, M.G. All authors have read and agreed to the published version of the manuscript.

**Funding:** The authors acknowledge the financial support from the Slovenian Research Agency (research core funding No. P2-0209 and P2-0095).

**Informed Consent Statement:** Informed consent was obtained from all subjects involved in the study.

**Data Availability Statement:** The data collected during this study are available on request from the corresponding author. The data are not publicly available due to privacy reasons.

**Acknowledgments:** The authors would like to thank Eva Černčič, Jernej Zupančič, Matej Trček, Tine Kolenik and Igor Gornik for their contributions.

**Conflicts of Interest:** The authors declare no conflict of interest.

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
