# Peer review of "A One-Dimensional Non-Intrusive and Privacy-Preserving Identification System for Households"

_electronics, doi:10.3390/electronics10050559_

Round 1

Reviewer 1 Report

  • Although the drawbacks of the existing methods have been discussed, the challenges of this work are suggested to be further clarified. The analysis of the extracted results should be extended.
  • The experimental results must be correlated with those obtained from the simulation, so that the research or study can be validated

Reviewer 2 Report

Paper deal with an interesting problem where mainly experimental investigation is presented but not well evaluated and compared to other solutions.

Paper has good potential, but current presentation in not in the level for this journal yet. SO please update at minimum:

  • Need of comparative evaluation is in these papers more than welcome to have a possibility to compare relevant data based on same data or same style of testing..
  • - comparison need to be in tables. TP, FP, etc.. and more detailed description.. including HW info.. epochs.. etc.
  • methodology need to be enhanced..
  • Conclusions need to be a separate section. contain PROS and CONS of your solution. Contribution.. and future direction..   

Reviewer 3 Report

This paper tests a sensor based on single dimension measurement for a novel approach to non-intrusive and privacy-preserving people identification in a smart environment, e.g., a household. Since the required accuracy in this AmI application is quite low compared to the security tasks, and since the number of people in a household is typically small, the proposed approach based only on body height or silhouette, e.g., body features, might prove sufficient for real-life applications.

However, there are several weakness and typos in the paper, which make the paper cannot be fully appreciated. The problems are identified as follows:

  1. As mentioned in this paper in line 517, "the classification accuracy decreases rapidly as the number of people increases, dropping to 90 percent for 18 people". The limitation of the number of people in experiments makes the less practical result.
  2. In this paper, comparison of different experiments is too less. The authors only compare the parameter of height and shape of bodies, which makes the result of experiment not valid and has no reference value.
  3. The authors fail to cite several prior literatures (e.g., [1-4]) highly related to it and comprehensively discuss the differences between them and this paper.

[1] TrueHeart: Continuous Authentication on Wrist-worn Wearables Using PPG-based Biometrics. INFOCOM 2020

[2] Incentive mechanism for privacy-aware data aggregation in mobile crowd sensing systems, in TON 2019.

[3] LPPA: Lightweight Privacy-Preserving Authentication From Efficient Multi-Key Secure Outsourced Computation for Location-Based Services in VANETs, in TIFS 2020.

[4] Enabling privacy-preserving incentives for mobile crowd sensing systems, in ICDCS  2016.

  1. The "privacy-preserving" theme is mentioned many times in this paper, but the author never states which kind of privacy-preserving standard they obey, and even not mention which aspects of privacy that has been preserved. It is only empty and vague concepts which is not scientistic and reasonable.
  2. The trial of different machine learning methods is lacking, which is needed to implemented in the further work.

Round 2

Reviewer 2 Report

Dear Authors,

Thanks for all the corrections. I am satisfied with the extent of the changes you have made to the manuscript. Overall, the quality of the article has increased significantly.

Authors incorporated my suggestion a most in the sense of updated measurementsintroduction, methodology and discussion.. 

Thus I suggest to accept article in present form.

Reviewer 3 Report

The authors have addressed my comments to the previous version. I do not have any other comment and recommend that this paper be accepted.